# Perception of Telepsychiatry in Saudi Adults with Major Depressive Disorder and Validation of the Telehealth Satisfaction Scale: A Cross-Sectional Study

**DOI:** 10.3390/healthcare13172149

**Published:** 2025-08-28

**Authors:** Musaab Alruhaily, Salman Althobaiti, Abdulmohsen Almutairi, Sami Al-Dubai, Ashaima’a Madkhali, Helal Alobaidi, Fahad Hameed Alharbi, Jalal Qasem Alziri

**Affiliations:** 1Preventive Medicine Post Graduate Studies, Al Madinah Health Cluster, Ministry of Health, Al Madinah 42210, Saudi Arabia; samidobaie@yahoo.com; 2Family and Community Medicine and Medical Education Department, Faculty of Medicine, Taibah University, Madinah 42353, Saudi Arabia; sthobaiti@taibahu.edu.sa; 3King Salman Bin Abdulaziz Medical City, Psychiatric Specialist Hospital, Al Madinah Health Cluster, Ministry of Health, Al Madinah 42210, Saudi Arabia; abdulmohsin.1989@gmail.com (A.A.); ashimaa_m@hotmail.com (A.M.); healobaidi@moh.gov.sa (H.A.); fahalharbi210353@moh.gov.sa (F.H.A.); jalziry@moh.gov.sa (J.Q.A.)

**Keywords:** telepsychiatry, depression, patient satisfaction, Saudi Arabia, TeSS, digital health

## Abstract

**Background:** Telepsychiatry expanded rapidly during the COVID-19 pandemic, yet patient experience data from mixed urban and rural areas in Saudi Arabia remain scarce. **Objective:** We aimed to quantify the perception of telepsychiatry among adults with major depressive disorder [MDD] in Madinah City, the KSA, and to identify associated demographic and clinical factors. **Methods:** A cross-sectional survey was conducted at Madinah Mental Health Hospital between December 2024 and March 2025. Eligible participants were Arabic-speaking adults [≥18 years] with a clinician-confirmed diagnosis of major depressive disorder [MDD] according to the Diagnostic and Statistical Manual of Mental Disorders, Fifth Edition [DSM-5], following a scheduled video- or audio-based telepsychiatry consultation. Perception of telepsychiatry was assessed using the validated 10-item Arabic version of the Telehealth Satisfaction Scale [TeSS], which evaluates audiovisual quality, communication, and support. Variables associated with perception at *p* < 0.20 in the bivariable analyses were entered into a multiple linear regression model to identify independent predictors. **Results:** Of the 218 eligible patients, 207 participated [response rate = 95%], with similarly high participation rates being reported in comparable telepsychiatry surveys [e.g., 90–91%]. The majority were male [59%], with a mean [SD] age of 38.4 [11.2] years. The mean satisfaction score was 32.3 ± 6.3, and 36% of participants were classified as highly satisfied. In the multivariable analysis, higher satisfaction was independently associated with male gender [B = 3.0, 95% CI: 1.3–4.7], intermediate versus elementary education [B = 4.3, 95% CI: 1.1–7.6], and the presence of a chronic illness [B = 2.1, 95% CI: 0.3–3.8]. **Conclusions:** Telepsychiatry is generally well-accepted among adults with depression in Madinah. However, women and individuals with lower educational attainment report lower satisfaction. Targeted interventions such as improving privacy, offering digital literacy support, and tailoring communication may help improve the telepsychiatry experience for underserved groups.

## 1. Introduction

The onset of the COVID-19 pandemic has dramatically altered the landscape of healthcare delivery worldwide, including in Saudi Arabia. As part of efforts to continue providing essential medical services while minimizing the risk of virus transmission, telemedicine has emerged as a critical tool in maintaining the continuity of care [1]. In Madinah, telemedicine has been utilized to support mental health services for patients with conditions such as depression, anxiety, and other psychiatric disorders. Telemedicine leverages advanced technologies to overcome geographical barriers and expand access to care. It allows for the remote diagnosis, treatment, and monitoring of patients, especially beneficial in a geographically diverse country like Saudi Arabia [2]. The convenience of telemedicine, including reduced travel and waiting times, presents a compelling alternative to traditional in-person consultations. This is particularly valuable for patients in remote or underserved areas who may otherwise encounter significant obstacles to accessing healthcare services [3]. Additionally, in conservative societies, such as parts of Saudi Arabia, telemedicine provides a discreet and less intimidating option for individuals who might fear social stigma associated with visiting a mental health facility. For some patients, the anonymity of virtual consultations may reduce anxiety and increase willingness to seek care [4]. Despite these clear advantages, a portion of patients still prefer in-person consultations. Factors contributing to this preference include the perceived superiority of direct physical examinations, stronger personal interaction with clinicians, and a more reassuring therapeutic experience. Many patients report feeling more understood and supported during face-to-face interactions, which can enhance trust and satisfaction [5]. Conversely, several studies report that telepsychiatry can foster a therapeutic alliance comparable to—or predictive of outcomes similarly to—face-to-face care, indicating that digital modalities may actively support, not only jeopardize, clinician–patient relationships [6].

These dynamics highlight the need for a deeper understanding of how patients perceive and respond to telepsychiatry services in contrast to traditional in-person care. Several studies in Saudi Arabia have examined patient satisfaction with telepsychiatry, often reporting favorable outcomes. For instance, Almalky et al. found that 94.3% of psychiatric patients at King Khalid University Hospital were satisfied with telepsychiatry, particularly in terms of privacy, comfort, and communication quality [7]. Gada et al. also reported positive perceptions in a cross-sectional study at King Abdulaziz Medical City, noting satisfaction with accessibility, effectiveness, and safety [8]. Similarly, a study by Magadmi and Kamel in Jeddah found high levels of patient satisfaction with teleconsultations during the COVID-19 pandemic, citing convenience and reduced stigma as key benefits [9]. Although these findings suggest a generally positive reception of telepsychiatry in Saudi Arabia, there is a lack of data specifically addressing the perspectives of patients in Madinah. Furthermore, few studies have compared the perceptions of telepsychiatry directly with those of in-person consultations within the same population. Therefore, the objective of this study is to evaluate the perspectives and satisfaction levels of mental health patients using telemedicine services in Madinah and to compare these perspectives with those related to traditional in-person psychiatric consultations.

## 2. Materials and Methods

### 2.1. Study Design and Ethics

We performed a hospital-based cross-sectional survey between 1 December 2024 and 31 March 2025 at Madinah Mental Health Hospital. This study was approved by the Madinah Regional Institutional Review Board [approval MMHH-2024-149] and adhered to the Strengthening the Reporting of Observational Studies in Epidemiology [STROBE] guideline [10]. All participants provided their electronic informed consent before answering the survey.

### 2.2. Participants

Eligible participants were Arabic-speaking adults [≥18 years old] with a clinician-confirmed diagnosis of major depressive disorder [MDD] according to the Diagnostic and Statistical Manual of Mental Disorders, Fifth Edition [DSM-5] [11], recorded in their electronic medical record, and who completed a scheduled video- or audio-based telepsychiatry visit during the study window. We excluded [i] patients in acute crisis requiring immediate in-person care and [ii] individuals with cognitive impairment that precluded reliable survey completion. Acute crisis was defined as active suicidal ideation with intent, psychotic agitation, or any presentation requiring same-day in-person assessment or admission. Of 218 eligible patients, 207 completed the questionnaire, yielding a 95% response rate.

### 2.3. Sample Size Calculation

Assuming a conservative prevalence of 50% high satisfaction, a sample of 384 patients would provide a ±5 percentage point margin of error at the 95% confidence level [*n* = 384 = 1.96^2^ × 0.5 × 0.5/0.05^2^]. The achieved sample of 207 yields a margin of ±6.8%. Post hoc, this sample affords 93% power to detect a 4-point difference in TeSS score between two equally sized groups [SD = 6, α = 0.05].

### 2.4. Study Instruments and Tools

The study instrument consisted of two main components. The first part collected sociodemographic and clinical information, including age, gender, marital status, highest level of education, monthly income, place of residence [urban vs. rural], number of years since depression diagnosis, presence of chronic medical conditions such as diabetes or hypertension, and the type of telepsychiatry consultation used [video vs. audio].

The second part assessed patient perception using the validated 10-item Telehealth Satisfaction Scale [TeSS]. This tool evaluates satisfaction with multiple aspects of telepsychiatry services and is widely used in healthcare research. Each item is rated on a 4-point Likert scale ranging from 1 [strongly disagree] to 4 [strongly agree], generating a total score between 10 and 40, with higher scores indicating greater satisfaction [11]. The items address key domains including audiovisual quality, personal comfort, ease of access, consultation duration, clarity of treatment explanation, clinician professionalism, respect, privacy, and responsiveness to patient concerns.

The 10 items are conceptually grouped into three domains: technical quality [items 1 and 2], interaction and care process [items 3 to 7], and communication and support [items 8 to 10]. For analysis, total TeSS scores were divided into tertiles to represent low [10–19], moderate [20–29], and high [30–40] satisfaction levels. The TeSS has demonstrated strong psychometric performance, including excellent internal consistency [Cronbach’s alpha = 0.83], and its validity has been confirmed in previous studies [9].

### 2.5. Validation of the Arabic Version of the TeSS

The English version of the Telehealth Satisfaction Scale [TeSS] was translated into Arabic following the World Health Organization’s five-step protocol to ensure linguistic and cultural validity [11]. First, two bilingual psychiatrists independently performed forward translations of the original scale, which were reconciled into a single unified Arabic version. Next, a professional medical linguist—blinded to the original English version—performed a back-translation to verify the semantic accuracy. An expert panel comprising two consultant psychiatrists and the linguist reviewed all versions of the scale to resolve discrepancies and confirm conceptual equivalence [11]. A pilot study was then conducted with 20 telepsychiatry patients to evaluate clarity and understandability, ensuring linguistic precision and content validity.

The Arabic TeSS demonstrated excellent internal consistency [Cronbach’s α = 0.90], with corrected item–total correlations of 0.58–0.69. The exploratory factor analysis supported a one-factor solution: KMO = 0.820; Bartlett’s χ^2^ = 591.41; df = 45; *p* < 0.001; eigenvalue = 5.09, accounting for 50.9% of variance. All items loaded ≥0.51 [range 0.51–0.74].

Although a single-item global satisfaction anchor was not collected, several findings support construct validity: [i] a strong single-factor structure with all loadings > 0.50; [ii] high internal consistency [α = 0.90]; and [iii] known groups validity, with the TeSS scores differing by gender, education, and chronic disease in theoretically expected directions. Future studies should add an external criterion [e.g., a global satisfaction item or intention-to-reuse] to more formally establish convergent validity.

### 2.6. Data Collection

Immediately after their telepsychiatry session, patients received a secure SMS link to the survey [Qualtrics platform]. Those preferring assistance completed the questionnaire by telephone with a trained research assistant.

### 2.7. Statistical Analysis

Statistical analyses were conducted using IBM SPSS Statistics version 30.0.0. Continuous variables were summarized as mean ± standard deviation [SD], while categorical variables were presented as counts and percentages. The normality of continuous data was assessed using the Shapiro–Wilk test.

For univariate analyses, independent-samples *t*-tests and a one-way ANOVA were performed to examine the differences in Telehealth Satisfaction Scale [TeSS] scores across demographic and clinical variables. Variables associated with TeSS scores at a significance level of *p* < 0.20 in univariate tests were subsequently included in a multivariable linear regression model. Variables with *p* < 0.20 in univariate analyses were entered into the multivariate model using backward stepwise elimination.

The final model used ordinary least squares with heteroscedasticity-consistent [HC3] standard errors. Multicollinearity was minimal [all VIF < 1.8]. Missing data were ≤2% per variable and handled with listwise deletion; Little’s MCAR test supported missing completely at random [*p* = 0.41]. Residual normality showed a small deviation [Shapiro–Wilk W = 0.964, *p* < 0.001], with skewness −0.420 and kurtosis −0.365. Given approximate symmetry and the use of HC3, bias was unlikely to meaningfully affect the inferences. Two-tailed *p* < 0.05 denoted statistical significance.

Although the TeSS score is a bounded outcome, OLS regression was selected for its simplicity and interpretability. The assumptions of linearity and independence were met; however, residual normality was slightly violated. The Shapiro–Wilk test indicated a statistically significant deviation from normality [W = 0.964, *p* < 0.001], but residual skewness [–0.420] and kurtosis [–0.365] were within the acceptable limits. The distribution was approximately symmetric, and robust standard errors were applied to mitigate heteroscedasticity. As such, the minor non-normality was not considered to meaningfully bias the results.

Future studies may consider beta or ordinal logistic regression to better accommodate the scale’s distributional properties.

## 3. Results

Table 1 shows that exploratory factor analysis supported a single-factor solution, consistent with the original validation. The first factor explained 50.9% of the variance, with an eigenvalue of 5.09, and all 10 items showed satisfactory loadings ranging from 0.51 to 0.74. The internal consistency was excellent [Cronbach’s α = 0.90], replicating the psychometric strength reported for the original TeSS.

Participant characteristics. The sample was mostly male and based in Madinah. About two-thirds reported a chronic condition. Most visits were new and conducted by video. Detailed counts and percentages are shown in Table 2.

### 3.1. Patient Satisfaction

Participant satisfaction was generally favorable across the items; privacy tended to score highest and image quality lowest (Table 3).

### 3.2. Factors Associated with Patient Satisfaction: Univariate Analysis

Univariate analysis. Higher satisfaction was observed for men, intermediate education, and new consultations; chronic disease showed a positive trend, while audio tended to be lower than video. See Table 4 for estimates and *p*-values.

Education displayed a nonlinear pattern with the highest scores at the intermediate level, consistent with a potential “optimal familiarity” effect (see Table 4).

### 3.3. Factors Associated with Patient Satisfaction in Multivariate Analysis

The results of the multivariable linear regression are shown in Table 5. The primary model included education, gender, and chronic disease [reference categories: elementary education, female, and no chronic disease]. This model explained R^2^ = 0.186 of variance [F [3,203] = 15.5, *p* < 0.001] with adjusted R^2^ = 0.168, indicating modest explanatory power.

To explore additional covariates, a comprehensive model included consultation type, education, age, gender, consultation modality, chronic disease, marital status, income, residence, and years since diagnosis. The overall fit was modest [R^2^ = 0.115; adjusted R^2^ = 0.070; F [10,196] = 2.56, *p* = 0.006], indicating limited incremental explanatory value beyond the parsimonious model.

Multivariable results. In the full model, male gender (B = 3.02; 95% CI 1.32–4.71; *p* = 0.001), intermediate education vs. elementary (B = 4.31; 1.05–7.56; *p* = 0.010), high school vs. elementary (B = 2.20; 0.48–3.92; *p* = 0.012), and chronic disease (B = 2.06; 0.31–3.80; *p* = 0.021) were independently associated with higher TeSS scores. Audio consultations trended lower than video (B = −3.37; *p* = 0.086). Multicollinearity was minimal (VIFs 1.01–1.11) (see Table 5).

## 4. Discussion

This study examined satisfaction with telepsychiatry among adults with major depressive disorder in the Madinah region and explored how satisfaction varied by patient characteristics. Overall, the participants reported high levels of satisfaction [mean TeSS = 32.3 ± 6.3], supporting the acceptability of remote psychiatric care in a mixed urban–rural Saudi Arabian setting [7].

### 4.1. Education and Digital Literacy

Participants with intermediate schooling reported the highest satisfaction, exceeding those with only elementary education by about five points and those with high school or university qualifications by smaller margins. This nonlinear pattern may indicate that moderate educational attainment provides sufficient digital literacy to navigate telehealth platforms, while higher education raises expectations that are harder to meet. Similar “inverted U-shape” effects have been reported in telemedicine studies from Europe, where mid-level digital familiarity was optimal. Training modules tailored to lower-educated patients and expectation management for higher-educated users may help to narrow these gaps.

Future service development could consider culturally informed strategies that support patient privacy without assuming uniform needs across demographic groups [12,13].

### 4.2. Chronic Disease and Engagement

Contrary to concerns that medically complex patients might struggle with remote interfaces, participants with coexisting chronic illnesses reported modestly yet significantly higher satisfaction. This supports prior findings that such patients—being more accustomed to frequent healthcare interactions—may adapt more readily to telehealth and value the convenience of remote consultations. These results underscore the importance of integrating telepsychiatry into care pathways for patients with multiple chronic conditions [14].

### 4.3. Non-Significant Factors

In adjusted analyses, income, place of residence, and consultation type [new vs. follow-up] were not independently associated with satisfaction, despite consultation type showing significance in the univariate tests [*p* = 0.004]. The lack of a rural–urban divide contrasts with early telemedicine concerns about connectivity; our findings likely reflect the recent expansion of national digital health infrastructure, even in villages. Similarly, income had no bearing on satisfaction, suggesting that subsidized national platforms mitigate financial barriers.

### 4.4. Strengths and Limitations

The strengths of this study include a focused sample of patients with a single diagnosis, the use of a rigorously translated and reliable satisfaction scale, and robust heteroscedasticity-consistent regression methods. The reduced sample size [207 vs. 384 target] lowers the precision of estimates [±6.8% vs. ±5% margin] and may limit power for subgroup comparisons. Several limitations warrant caution. First, the cross-sectional design precludes causal inference, and unmeasured factors [e.g., depression severity and clinician communication skills] may confound associations. Second, reliance on self-report immediately post-consultation could bias the ratings upward through recall or social-desirability effects. Third, the single-center setting may limit generalizability beyond Madinah. While both urban and rural participants were included, no formal subgroup analyses were conducted; thus, representativeness outside this context remains uncertain. Although a single-item global satisfaction measure was not included, the TeSS demonstrated strong internal consistency [α = 0.90] and a clear one-factor structure [all loadings > 0.50], supporting its construct validity. Future research should incorporate a global satisfaction anchor or an external scale to formally assess convergent validity.

### 4.5. Sample Size and Generalizability

Although our initial sample size calculation recommended 384 participants to achieve ±5% precision, only 207 were enrolled, yielding a ±6.8% margin. This reduced sample size likely limited the statistical power, particularly for subgroup comparisons, increasing the risk of type II error for small effects. Additionally, excluding patients in acute psychiatric crisis—while ethically necessary—may bias the findings toward participants with more stable or milder symptoms. Consequently, generalizability to adults with major depressive disorder requiring urgent in-person care is limited. This single-center design further constrains external validity beyond the Madinah region. Future multi-center studies with larger, more diverse samples, including acute cases where feasible, are warranted.

### 4.6. Implications and Future Directions

Our findings affirm that telepsychiatry is broadly acceptable to adults with depression in Saudi Arabia. To optimize care, services should incorporate culturally sensitive practices that emphasize privacy and confidentiality, particularly in contexts where stigma may influence willingness to seek help. In our data, gender and chronic disease status were not significant predictors of privacy-related ratings; therefore, any subgroup-targeted privacy strategies should be considered exploratory and evaluated in future research. While concerns about reduced personal connection in telepsychiatry are valid, emerging evidence suggests that therapeutic alliance can be maintained—and even strengthened—via digital platforms. Some patients may feel more comfortable disclosing sensitive information from home, potentially enhancing openness and rapport [15]. Flexibility, reduced travel burden, and the option to connect remotely may improve engagement and continuity. Evidence from digital mental health interventions [e.g., individuals diagnosed with schizophrenia] indicates that strong therapeutic relationships can be developed through virtual care. These findings underscore the importance of clinician training in digital empathy and telepresence [15].

Additional strategies might include tiered educational supports to bolster digital literacy, and targeted outreach to patients new to telehealth; however, these recommendations are proposed as best-practice considerations rather than conclusions drawn from subgroup effects in our sample. Future studies should longitudinally assess whether satisfaction predicts adherence and clinical outcomes and the test interventions aimed at addressing patient needs identified here. We did not measure depression severity or clinician communication style, which may confound satisfaction.

## 5. Conclusions

Telepsychiatry is a well-received modality for delivering care to adults with depression in the Madinah region, with the majority of patients reporting moderate to high satisfaction. Notably, men, individuals with intermediate educational attainment, and those managing chronic medical conditions alongside depression demonstrated the highest satisfaction levels. These insights highlight the importance of tailoring telepsychiatry services by enhancing privacy and user support for women, offering digital literacy resources for varying education levels, and integrating telehealth into chronic care pathways to maximize patient engagement and outcomes. As telepsychiatry becomes an enduring component of mental health infrastructure, such targeted strategies will be critical to ensuring equitable, high-quality care across diverse patient groups.

## Figures and Tables

**Table 1 healthcare-13-02149-t001:** Factor loadings for the 10-item Arabic Telehealth Satisfaction Scale [TeSS] and summary psychometric statistics.

Item Statement	Factor 1 Loading
Was your privacy respected during the session?	0.7
Did the clinic team answer your questions and clarify your inquiries about how to communicate with them remotely?	0.7
How do you rate the doctor’s attention to your condition?	0.7
How do you rate the image quality during remote communication with the doctor?	0.6
Was the duration of the session with the doctor sufficient?	0.6
How respectful was the medical team in their interaction with you?	0.6
Did the doctor explain the treatment clearly?	0.6
How comfortable did you feel communicating with the doctor remotely?	0.6
How easy was it to access the doctor remotely?	0.5
How do you rate the sound quality during remote communication with the doctor?	0.5

**Table 2 healthcare-13-02149-t002:** Sociodemographic and health characteristics of the study participants.

Characteristic	Category	Freq	%
**Age**	19–30	49	23.6
	31–40	35	16.9
	41–50	48	23.1
	51–60	47	22.7
	61–70	28	13.5
**Gender**	Male	123	59.4
	Female	84	40.6
**Income**	No Income	16	7.7
	SAR ≤ 5000	63	30.4
	SAR 5001–10,000	78	37.7
	SAR 10,001–20,000	50	24.2
**Marital Status**	Single	72	34.8
	Married	77	37.2
	Divorced/Widowed	58	28
**Education**	Elementary	32	15.5
	Intermediate	16	7.7
	High School	98	47.3
	University	61	29.5
**Residence**	Madinah	183	88.4
	Another City	10	4.8
	Village	14	6.8
**Chronic Disease**	Yes	135	65.2
	No	72	34.8
**Consultation Type**	Follow-up	80	38.6
	New	127	61.4
**Consultation Modality**	Video	196	94.7
	Audio	11	5.3
**Years Since Diagnosis**	<3	111	53.6
	>3	96	46.3

**Table 3 healthcare-13-02149-t003:** Participants’ satisfaction ratings with remote doctor consultations.

Statement	Very Dissatisfied	Dissatisfied	Satisfied	Very Satisfied	Mean ± SD
How do you rate the sound quality during remote communication with the doctor?	19	24	45	119	3.2 ± 0.9
How do you rate the image quality during remote communication with the doctor?	27	30	45	105	3.1 ± 1.0
How comfortable did you feel communicating with the doctor remotely?	20	23	48	116	3.2 ± 0.9
How easy was it to access the doctor remotely?	21	32	45	109	3.1 ± 1.0
Was the duration of the session with the doctor sufficient?	21	28	51	107	3.1 ± 1.0
Did the doctor explain the treatment clearly?	18	31	56	102	3.1 ± 0.9
How do you rate the doctor’s attention to your condition?	18	30	50	109	3.2 ± 0.9
How respectful was the medical team in their interaction with you?	18	23	46	120	3.2 ± 0.9
Was your privacy respected during the session?	18	17	49	123	3.3 ± 0.9
Did the clinic team answer your questions and clarify your inquiries about how to communicate with them remotely?	19	21	50	117	3.2 ± 0.9

**Table 4 healthcare-13-02149-t004:** Comparison of mean satisfaction scores by participant characteristics with *p*-values.

Characteristic	Category	Mean	SD	*p*-Value
**Age**	19–30	36.0	4.6	0.371
	31–40	32.4	5.5	
	41–50	33.0	6.5	
	51–60	31.8	6.8	
	61–70	32.4	6.0	
**Gender**	Male	33.3	6.2	0.003
	Female	30.6	6.1	
**Income**	No Income	33.3	8.5	0.501
	SAR < 5000	32.2	6.0	
	SAR 5000–10,000	32.7	6.6	
	SAR 10,001–20,000	31.1	5.4	
**Marital Status**	Single	32.4	7.1	0.112
	Married	33.1	6.1	
	Divorced/Widowed	30.8	5.3	
**Education**	Elementary	30	6.5	0.019
	Intermediate	34.8	7.1	
	High School	33.1	6.0	
	University	31.3	6.1	
**Resident**	Madinah	32.1	6.1	0.732
	Another City	33.5	8.7	
	Village	33.0	6.3	
**Chronic Disease**	Yes	33.0	6.4	0.053
	No	30.8	5.9	
**Consultation Type**	Follow-up	31.3	5.4	0.004
	New	32.8	6.7	
**Consultation Modality**	Video	32.4	6.3	0.188
	Audio	29.8	5.5	
**Years Since Diagnosis**	<3	33.0	5.8	0.099
	>3	31.5	6.7	

**Table 5 healthcare-13-02149-t005:** Multivariable linear regression of participant characteristics predicting satisfaction scores.

Variable		B	CI	T	*p*-Value	VIF
Education	Elementary	**Reference**				
	Intermediate	4.3	1.0–7.5	0.8	0.010	1.1
	High School	2.2	0.4–3.9	0.9	0.012	1.0
Gender	Female	**Reference**				
	Male	3.0	1.3–4.7	0.9	0.001	1.0
Chronic disease	No	**Reference**				
	Yes	2.0	0.3–3.8	0.9	0.021	1.01

## Data Availability

The datasets generated and/or analyzed during the current study are available from the corresponding author upon reasonable request.

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
