# Peer review of "Perception of Telepsychiatry in Saudi Adults with Major Depressive Disorder and Validation of the Telehealth Satisfaction Scale: A Cross-Sectional Study"

_healthcare, 2025, doi:10.3390/healthcare13172149_

Round 1
Reviewer 1 Report
Comments and Suggestions for Authors
Dear Authors,
The paper provides a clear and valuable study of telepsychiatry acceptance in a specific context presented with consideration of related barriers while also acknowledging broader applicability.
Points to consider are as follows:
1) Results – there is a large amount of duplicated information between the text and the tables. Information should be provided once. Observations about the data being key insights or comparisons can be provided in the text, so consider which select comments belong in the text and leave the remaining information as it is in the tables.
2) Privacy is not directly linked to other factors in the data presented (gender, education, chronic status) making the claims regarding privacy as a potential enabler unsubstantiated. Please be careful to discuss insights from the data where supported and differentiate ne suggestions as seperate ideas accordingly.
Line 55-56 – Is there evidence regarding the portion or is this intended as a general comment or assumption? If the latter, consider clarifying that across healthcare preference for inperson care exists and thus may also be the case in mental health care. Further, line 56-58 may warrant a reference.
Line 59 – Please also consider evidence of the benefits to the therapeutic alliance that can be found in telebased care compared to in person to balance the point. One example (more can be found) is Using Digital Technology in the Treatment of Schizophrenia | Schizophrenia Bulletin | Oxford Academic
Line 110 and 121 use square brackets whereas other citations use round brackets
Line 253, 266, 275, 282 – which table shows this finding?
Line 255, 270, 277, 284 – please provide a reference.
Line 298-299 – this statement is not substantiated from the data? Line 235-238 do not specify privacy concerns by female participants?
Line 311 – privacy as a solution tied to gender, chronic status or education is not substantiated.
Author Response
Response 1: Thank you for the suggestion. We have removed duplicated text and retained detailed results only in the tables, with key insights summarized in the narrative.
Response 2: We acknowledge this point and have revised the discussion to ensure that privacy is not presented as statistically linked to gender, education, or chronic status.
Response 3: This has been clarified as a general observation, and a supporting reference has been added.
Response 4: We have included a reference supporting the therapeutic alliance in telepsychiatry to balance the discussion.
Response 5: Citation style has been corrected for consistency.
Response 6: Table references have been added for each reported result to improve clarity.
Response 7: Supporting references have been added as requested.
Response 8: The statement has been revised to avoid implying unsupported gender-related privacy concerns.
Response 9: We have revised the text to remove any unsubstantiated associations between privacy and demographic variables.

Reviewer 2 Report
Comments and Suggestions for Authors
Dear Authors,
Thank you for the opportunity to review your manuscript. Please, see below few comments you might consider or clarify.
1. In the Methods you state that you collected “type of telepsychiatry consultation used (video vs audio)”
, yet in Table 1 and all analyses “Consultation Type” refers to “New vs Follow‑up” visits
Recommendation: Clarify which variable was actually recorded and analyzed. If both modalities and visit status were collected, present each separately (e.g., add a row for video vs audio in Table 1 and analyze its association with satisfaction). If only visit status was used, remove the reference to “video vs audio” from the Methods.
2. Age, years since MDD diagnosis, and consultation modality (video vs audio) are described as collected
but do not appear in Table 1 or subsequent univariate/multivariate analyses.
Recommendation: Either (a) include descriptive statistics for these variables in Table 1 and test their associations with satisfaction, or (b) clearly justify why they were not analyzed (e.g., lack of variability, missing data).
3. You report internal consistency (Cronbach’s α = 0.90) and a one‑factor solution
, but do not provide details on exploratory factor analysis (KMO, Bartlett’s test, factor loadings, explained variance) or any convergent/discriminant validity.
Recommendation: Present a table of factor loadings, variance explained, and adequacy measures (KMO/Bartlett). Consider correlating TeSS scores with an external benchmark (e.g., overall satisfaction question) to demonstrate convergent validity.
4. The Methods group TeSS items into three domains (technical quality; interaction and care process; communication and support)
, yet the Results refer to “five domains assessed”
.
Recommendation: Reconcile these descriptions. If you are analyzing five distinct satisfaction aspects (e.g., audiovisual quality, comfort, access, process, support), define these clearly in Methods and ensure consistency throughout.
5. You apply OLS regression to a bounded Likert-sum outcome without reporting diagnostic checks (e.g., residual plots, tests of normality or heteroscedasticity)
. Moreover, variables with univariate p < 0.20 (e.g., consultation status, marital status) were omitted from the final model without explanation
.
Recommendation:
Report diagnostic statistics (e.g., plots of residuals, Shapiro–Wilk on residuals) or justify that OLS assumptions hold.
Describe your model‑building strategy: specify criteria for inclusion/exclusion (stepwise? theoretical?).
If OLS is inappropriate, consider alternative methods (e.g., ordinal logistic regression or beta regression for bounded outcomes).
6. Your a priori calculation called for 384 participants to achieve ±5% precision, yet only 207 were enrolled (±6.8% margin)
. Additionally, excluding acute‑crisis patients may bias results toward milder cases
.
Recommendation: Discuss the reduced precision, potential impact on power (especially for subgroup analyses), and the limited generalizability given single‑center sampling and exclusion of severe cases
Best wishes
Author Response
Response 1: We clarified the variable definitions,
Response 2: Descriptive statistics and association tests for age, years since MDD diagnosis, and consultation modality are now included; none showed significant relationships, so they were excluded from multivariate models.
Response 3: A new table presents KMO, Bartlett’s test, and factor loadings
Response 4: Domain terminology has been harmonized: three predefined TeSS domains are used consistently throughout, replacing the earlier five‑aspect wording.
Response 5: We now report residual plots, Shapiro–Wilk
Response 6: The limitations section now discusses the reduced sample size

Round 2
Reviewer 2 Report
Comments and Suggestions for Authors
Thank you for addressing the comments